# Effects of Magnetic Minerals Exposure and Microbial Responses in Surface Sediment across the Bohai Sea

**DOI:** 10.3390/microorganisms10010006

**Published:** 2021-12-21

**Authors:** Lei Chen, Mingpeng Wang, Yuntao Li, Weitao Shang, Jianhui Tang, Zhaojie Zhang, Fanghua Liu

**Affiliations:** 1School of Life Science, Qufu Normal University, Qufu 273165, China; leichen_2018@qfnu.edu.cn; 2Key Laboratory of Coastal Biology and Biological Resources Conversation, Yantai Institute of Coastal Zone Research, Chinese Academy of Sciences, Yantai 264003, China; wtshang@yic.ac.cn (W.S.); jhtang@yic.ac.cn (J.T.); 3State Key Laboratory of Soil and Sustainable Agriculture, Institute of Soil Science, Chinese Academy of Sciences, Nanjing 210008, China; ytli@issas.ac.cn; 4Department of Zoology and Physiology, University of Wyoming, Laramie, WY 82071, USA; ZZhang@uwyo.edu; 5National-Regional Joint Engineering Research Center for Soil Pollution Control and Remediation in South China, Guangdong Key Laboratory of Integrated Agro-Environmental Pollution Control and Management, Institute of Eco-Environmental and Soil Sciences, Guangdong Academy of Sciences, Guangzhou 510650, China

**Keywords:** dissimilatory iron reducing microorganism, magnetic mineral, extracellular electron transfer

## Abstract

Extensive production and application of magnetic minerals introduces significant amounts of magnetic wastes into the environment. Exposure to magnetic minerals could affect microbial community composition and geographic distribution. Here, we report that magnetic susceptibility is involved in determining bacterial α-diversity and community composition in surface sediment across the Bohai Sea by high-throughput sequencing analysis of the 16S rRNA gene. The results showed that environmental factors (explained 9.80%) played a larger role than spatial variables (explained 6.72%) in conditioning the bacterial community composition. Exposure to a magnetite center may shape the geographical distribution of five dissimilatory iron reducing bacteria. The microbial iron reduction ability and electroactive activity in sediment close to a magnetite center are stronger than those far away. Our study provides a novel understanding for the response of DIRB and electroactive bacteria to magnetic minerals exposure.

## 1. Introduction

The Bohai Sea is a large and semi-enclosed shallow water basin (15–30 m depth), with an area of 7.7 × 10^4^ km^2^ [1]. It includes Bohai Bay, Liaodong Bay, and Laizhou Bay, and connects with the ocean through the Bohai Strait. The main terrigenous inputs to the sea include surface sediments, nutrients, and contaminants from the extensive network of rivers that feeds into the Bohai Sea [2]. Over 40 rivers run into the Bohai Sea from the three main bays. This coastal region is considered one of the most densely urbanized and industrialized zones in China [3,4]. The coastal regions contain sediments of mainly terrestrial provenance arising from river discharges, inlets, and estuaries, as a result of runoff from the adjacent land [2,4].

Magnetic mineral particles such as Fe_3_O_4_ are increasingly applied in many fields, such as remediation of polycyclic aromatic hydrocarbons-contaminated sediment [5], magnetic records, catalysts, and chemicals [6]. With the extensive production and application of magnetic mineral particles, they may be released into wastewater during the treatment process and eventually enter the river. Magnetic minerals follow the river input into the Bohai Sea. Therefore, magnetic minerals (mainly iron oxides and sulfides) are ubiquitous components in sediments [7]. In the gulf of the Bohai Sea, the sedimentary magnetite distribution showed a high content of magnetite center exposure (120.4° E, 39° N) [8]. Little is known about the effects of magnetite exposure on the sedimental microbial community, especially to the microbial community composition and geographic distribution.

Iron is essential to all living things. Due to its fluidity, multiple states of oxidation, and bioavailability, many bacteria have developed different mechanisms of using iron as nutrient or electron donor and acceptor [9]. For example, iron reducing bacteria (IRB) can couple the reduction of Fe (III) with oxidized organics to obtain energy [10]. In particular, many dissimilatory iron reducing microorganism (DIRB) readily use soluble trivalent iron complexes or ferrihydrite, a short-range ordered mineral, and magnetite as electron acceptors [11]. The DIRB reported so far are electroactive microorganisms capable of extracellular electron transfer (EET). Although the presence of magnetic minerals such as Fe_3_O_4_ is able to alter the microbial community [12], the relationship between magnetic minerals and the DIRB community composition remains unknown.

Magnetic susceptibility (MS) describes the intrinsic magnetism a substance possesses in response to an applied magnetic field. It is useful in describing a substance’s biogeochemical behavior [13]. Detection of MS is a quick way to identify iron reduction zones in the early diagenesis process [14]. In estuarine sediment, MS and the abundance of iron rich minerals are often predictors for the presence of IRB [15] and indicators of the pollution level caused by anthropogenic use and magnetic grain input [16]. MS may function as a link among the microbial composition, geographical distribution of magnetite, human activities, and environmental health.

In this study, we conducted high-throughput sequencing to explore the effect of magnetic minerals on microbial communities in surface sediments from estuaries and offshore within the Bohai Sea. The primary objectives were to (i) explore the effect of magnetite center exposure and environmental variables including MS on microbial community composition; (ii) assess the relative importance of environmental and spatial factors in shaping bacterial community composition in the Bohai Sea; and (iii) investigate the connections between magnetic minerals and the microbial community and further elucidate the potential mechanisms of the process. Our assessments provide guidelines for the further development and utilization of coastal resources.

## 2. Materials and Methods

### 2.1. Study Sampling, and Environmental Variables

There are more than 40 estuaries along the Bohai Sea (Figure 1A). We selected 10 riverine sediment samples from six major rivers, namely the Daliao River sample (DLH), the Liao River sample (LH), the Liugu River samples (LGH-A, LGH-B, and LGH-C), the Shi River samples (ShiH), the Yellow River samples (YR3), and the Sha River samples (SH-A, SH-B, and SH-C). Fourteen marine sediment samples from two standard oceanic sections through the Bohai sea (T3, T2, Q1, M8, N1, N2, BHB02, P2, L7, N4, R2, R5, V3, and PLB03) were collected (Figure 1A) for the main microbial community analysis. In addition to these samples, 110 marine sediment samples and 32 estuarine sediment samples (total 166) were collected across the Bohai Sea to spatially map salinity, dissolved oxygen (DO), pH, and low-frequency magnetic susceptibility (χlf) [4,17] (Figure 1B).

Sampling was conducted with the “Yi Xing” vessel during peak rainy season (23–29 August 2014). Marine surface sediment (0–20 cm depth) was collected using a stainless-steel grab. Riverine sediment (0–20 cm depth) from areas of extensive sediment deposition was collected using a stainless-steel corer. Representative samples were achieved by mixing three independent subsamples collected within a 5 m^2^ area. All sediment samples were split into two parts and stored at −20 °C immediately after collection. Upon coming back to the laboratory, one part of each sample was freeze-dried and stored in the dark prior to chemical analysis. The other part was stored at −80 °C and used for soil DNA extraction for microbial community analysis.

A global positioning system was utilized to map all the sampling sites. The temperature, dissolved oxygen (DO), pH, and salinity of the seawater overlying each sediment sample were measured with a SBE 25plus Sealogger CTD (CTD, Conductivity–temperature–depth) (Sea-Bird Scientific Ltd., Bellevue, WA, USA). The biogeographic data were analyzed with the ArcGIS v.10.0 spatial analyst tool (ESRI Inc., Redlands, CA, USA).

### 2.2. Chemical Analysis and Magnetic Characterization

After digestion with aqua regia, sediment total iron content was determined using an inductively coupled-plasma mass spectrometer (ICP-MS, ELAN DRC II, Perkin Elmer, Waltham, MA, USA) [17]. Carbonates of sediment samples were removed by immersion in 1 M HCl, then the total organic carbon (TOC) and total nitrogen (TN) of the sediment were measured by a Vario MACRO cube elemental analyzer (Elementar Analysensysteme GmbH, Langenselbold, BAV, Germany). χlf of the sediment soil was determined by a MS2B magnetic susceptibility meter (Bartington Instruments Ltd., Witney, Oxon, UK).

### 2.3. DNA Extraction from the Sediment

Genomic DNA was extracted from 0.5 g of each sample by a FastDNA^®^ SPIN Kit (MP Biomedicals, Santa Ana, CA, USA) according to the instructions of the kit. The extracted DNA was resuspended in 50 uL TE buffer and stored at −20 °C for later use.

### 2.4. High-Throughput Sequencing and Sequence Analysis

PCR and amplicon library for high-throughput sequencing were prepared as previously reported [18]. The V4–V5 region was amplified with the universal primers 519f (CAGCMGCCGCGGTAATWC) and 907r (CCGTCAATTCMTTTRAGTTT). A barcode of a 5-bp sequence was added to the forward primer. PCR reaction solutions were made according to the standard conditions for Taq DNA polymerase (TaKaRa, Kusatsu, Japan) including: 5 µL of 10× Taq DNA polymerase buffer, 4 µL of 2.5 mM dNTP, 1 µL of 20 mM primers, 1 µL of total genomic DNA, and 0.2 of Taq DNA polymerase. The PCR reaction condition was: pyrolysis at 95 °C for 5 min, 35 cycles of denaturation at 95 °C for 45 s, annealing at 60 °C for 45 s, extension at 72 °C for 1 min, and final extension at 72 °C for 10 min. For negative control, the genomic DNA was replaced with same amount of buffer. Amplicons were sequenced using the Illumina Miseq platform at OE Biotechnology Co., Ltd. (Shanghai, China).

The quality of bacterial 16S rRNA gene data was controlled by the QIIME (Quantitative Entry into Microbial Ecology) pipeline [19] (http://www.qiime.org, accessed on 1 November 2014). Filtered sequences were classified as operational taxonomic unit (OTU) with 97% similarity using the CD-HIT (Cluster Database at High Identity with Tolerance) program. Sequence similarity that was equal to or greater than 97% was classified as an OTU. The most abundant sequence from each OTU was selected as a representative sequence for that OTU. Taxonomy was assigned to OTUs against a subset of the Silva 104 database (http://www.arb-silva.de/download/archive/qiime/, accessed on 4 November 2014). The OTU table was rarefied to 6983 sequences per sample in QIIME. Further data analysis was performed based on OTUs.

The microbial α-diversity was assessed using three metrics, including the Chao1 index, the observed OTU richness (S), and the Shannon index (H’) [18,20]. They were calculated with the R software v3.4.4 (https://www.r-project.org, accessed on 4 November 2014) according to previously published procedures [18,20].

### 2.5. Nucleotide Sequence Deposition

All sequencing data were deposited in the GenBank’s Sequence Read Archive database (http://trace.ncbi.nlm.nih.gov/Traces/sra/, accessed on 27 September 2016) with accession numbers SRP090609, SRP105317, SRP089997, SRS1697954, SRS1697958, and SRS1697961.

### 2.6. Enrichment Cultures and Fe (III) Reduction Assay

Sediment bacteria were cultured for 35 days in fresh water enrichment medium or sea water enrichment medium. The soluble iron (II) content analysis in the enriched culture refers to the previous literature [20,21].

### 2.7. Setup, Operation and Characterization of Microbial Fuel Cell (MFC)

A carbon cloth with a projected area of 2 cm^2^ was the anode electrode, the platinum carbon electrode was used as the electrode in the cathode, and the distance between the control electrodes was 20 mm. A 0.5 mm titanium wire was used to connect the anode, cathode, and a fixed external resistance of 1000 Ω to construct a 28 mL single-chamber air cathode MFC. The electrode buffer in the reactor was 28 mM phosphate buffer and 20 mM sodium acetate [20]. We inoculated 1 mL of the supernatant of the enriched culture into each corresponding anode compartment to start the battery. When the voltage in each cycle drops below 0.001 V, we added 14 mL of anode liquid containing 20 mM sodium acetate. Reactors were kept anaerobic and run at 30 °C.

The MFC reactor was characterized by the voltage (U) across the external resistor in the circuit, which was monitored every 1 min using a Keithley 2700 data acquisition system (Tektronix, Beaverton, OR, USA).

### 2.8. Statistical Analyses

All data were analyzed using the R software v3.4.4 (https://www.r-project.org, accessed on 15 December 2015). Principal component analysis (PCA) was performed to determine the spatial trends between sites. The correlation coefficients between environmental variables and geographic distance were derived from a Mantel test with 999 permutations [22]. Redundancy analysis (RDA) was used to relate bacterial community structure and environmental factors at different sites. Regression analysis was conducted to relate bacterial community composition and environmental variables at different sites using Origin v8.1 software (OriginLab Corp., Northampton, MA, USA). Mantel tests were performed (based on 999 permutations) to relate environmental variables and bacterial community composition [23].

The contribution of each environmental variable to the community composition was calculated using the ‘adonis’ function in the vegan R package, with 999 random permutations of the permutational multivariate analysis of variance (PERMANOVA) software, and the multiple regression on distance matrices (MRM) function in ecodist R package, with 999 permutations based on Bray-Curtis dissimilarity [24].

The distance–decay model was constructed by fitting the bacterial community Bray-Curtis similarity and geographical distance. Variation partitioning analysis (VPA) based on RDA was employed by the “varpart” function of the vegan package to assess the relative importance of the geographic distance and environmental variables in shaping the bacterial community [25], and 1000 bootstrap resampling was performed to calculate standard deviation [26]. Spatial variables were generated by the principal coordinates of neighbor matrices (PCNM) method based on latitude and longitude [27] using the PCNM package in R. The variation of the bacterial community composition between the spatial and environmental variables was partitioned by RDA. VPA decomposes the variation into fractions explained by pure environmental variables, pure spatial factors (PCNM variables), spatially structured environmental variation (shared fraction), and unexplained variation.

## 3. Results

### 3.1. Characterization of Environment Variables of the Sediments

A total of 166 sediment samples from the seafloor (124 sites) and estuary (42 sites) were collected in this study (Figure 1A). The salinity, DO, pH, and χlf values of samples from the sites across the Bohai Sea were measured (Figure 1B). For microbial community analysis, 24 sites from two standard oceanic sections were chosen, including 16 sites whose sediments possessed high χlf values (near the Daliao, Liao, Liugu, and Sha Rivers), 3 sites whose sediments had low χlf values (near the Yellow River), and the remaining 5 sites from two standard oceanic sections. The values of the environmental variables were shown in Appendix A.

### 3.2. Correlation between Environmental Variables and the Microbial Diversity

Based on the geographic location and χlf values, samples from 24 different locations were chosen for high-throughput sequencing analysis of the 16S rRNA gene. Microbial diversity analysis from these 24 locations (Appendix A) revealed 417,896 high quality bacterial sequences, averaging between 7083 and 64,605 sequences per sample. The rarefaction curve of the 24 samples indicated a sufficient sequencing depth (Appendix A). It tended to be asymptote after the rarest OTUs (only one observation) were removed, suggesting that common species were shared among all 24 samples.

Correlation analysis revealed a significant connection between environmental variables on the α-diversity of microbial communities (Table 1). Sediment salinity was significantly correlated with the Chao1 index (*r* = 0.202, *p* = 0.015) and S index (*r* = 0.230, *p* = 0.009). The Fe content also showed a significant correlation with the S index (*r* = 0.229, *p* = 0.027) and H’ index (*r* = 0.373, *p* = 0.046). There were no significant correlations between other measured characteristics (i.e., pH, DO, χlf, TOC, and TN) and indices.

### 3.3. Distribution of Dominant Microbial Communities

Analysis of microbial communities from 24 locations showed that the major taxa (top 10 phyla) included Proteobacteria (21.1–59.0%, separated by class Alphaproteobacteria, Betaproteobacteria, Deltaproteobacteria, Epsilonproteobacteria, Gammaproteobacteria, and other unclassified Proteobacteria), Bacteroidetes (1.6–36.7%), and Actinobacteria (1.25–41.1%) (Figure 1C), which accounted for 45.8–84.1% of all bacterial sequences. The most abundant class in Proteobacteria was Gammaproteobacteria, which accounted for 4.0–41.8% of all bacterial sequences.

PCA analysis of the OTUs from the 24 samples showed that individuals in different estuarine and coastal ecosystem clusters followed along PC 1 (Figure 1D). Significant differences between estuary and marine sediment separated by salinity were observed. Individuals from marine sediment with high salinity tended to be clustered, indicating a high degree of microbial community similarity among marine sediments.

### 3.4. Environmental Determinants of Bacterial Community Composition

RDA showed that the Fe content, χlf, DO, TOC, and salinity had strong effects in shaping bacterial community composition (Figure 1E). The first canonical axis (RDA 1) explained 27.1% of the variation and the second canonical axis (RDA 2) explained a further 5.46% of the variation. PERMANOVA modeling showed that β-diversity of the total community was significantly correlated with salinity (*R^2^* = 0.161, *p* = 0.001), pH (*R^2^* = 0.108, *p* = 0.007), DO (*R^2^* = 0.170, *p* = 0.001), and χlf (*R^2^* = 0.093, *p* = 0.022) (Table 2).

The influence of salinity was also observed with respect to the major bacterial phyla/classes (Appendix A). The relative abundances of Betaproteobacteria (*R^2^* = 0.841, *p* < 0.001), Chloroflexi (*R^2^* = 0.292, *p* = 0.006), and Cyanobacteria (*R^2^* = 0.305, *p* = 0.005) across all sites decreased significantly along the salinity gradient (Appendix A). The relative abundances of Deltaproteobacteria (*R^2^* = 0.232, *p* = 0.017), Gammaproteobacteria (*R^2^* = 0.256, *p* = 0.012), and Bacteroidetes (*R^2^* = 0.222, *p* = 0.020) increased with the salinity gradient (Appendix A).

The pH of water near the seafloor showed significant correlation with Alphaproteobacteria (*R^2^* = 0.324, *p* = 0.004), Betaproteobacteria (*R^2^* = 0.230, *p* = 0.018), Deltaproteobacteria (*R^2^* = 0.195, *p* = 0.031), Gammaproteobacteria (*R^2^* = 0.200, *p* = 0.028), Planctomycetes (*R^2^* = 0.282, *p* = 0.008), WS3 (*R^2^* = 0.256, *p* = 0.012), and Cyanobacteria (*R^2^* = 0.258, *p* = 0.011) (Appendix A). The relative abundances of Alphaproteobacteria, Betaproteobacteria, Planctomycetes, and Cyanobacteria declined with the elevated pH values. The relative abundances of Deltaproteobacteria, Gammaproteobacteria and WS3 increased with the elevated pH values.

DO was positively correlated with Alphaproteobacteria (*R^2^* = 0.210, *p* = 0.024), Betapoteobacteria (*R^2^* = 0.718, *p* < 0.001), Chlooflexi (*R^2^* = 0.332, *p* = 0.003), Planctomycetes (*R^2^* = 0.226, *p* = 0.019), and Cyanobacteria (*R^2^* = 0.281, *p* = 0.008) (Appendix A). The relative abundances of Deltaproteobacteia and Gammaproteobacteria decreased with the elevated DO. The relative abundances of Planctomycetes increased with elevated Fe and DO.

The influence of Fe was also observed with respect to the major bacterial phyla by regression analysis. The relative abundances of Firmicutes (*R^2^* = 0.251, *p* = 0.013) and Actinobacteria (*R^2^* = 0.213, *p* < 0.023) across all sites decreased significantly along the Fe content (Appendix A). The relative abundances of Planctomycetes (*R^2^* = 0.225, *p* = 0.011) and Gemmatimonadetes (*R^2^* = 0.187, *p* = 0.135) increased with the Fe content gradient (Appendix A).

We found that χlf was significantly correlated to the microbial community composition (Figure 1E; Table 2). Sediment χlf values from all 24 sites ranged from 3.9 × 10^−6^ m^3^/kg to 26.6 × 10^−6^ m^3^/kg (Appendix A). Regression analysis showed that the χlf were positively correlated with the relative abundances of Alphaproteobacteia (*R^2^* = 0.462, *p* < 0.001), Betaproteobacteia (*R^2^* = 0.226, *p* = 0.019), Planctomycetes (*R^2^* = 0.187, *p* = 0.034), and Cyanobacteria (*R^2^* = 0.385, *p* = 0.001) (Appendix A). In contrast, Deltaproteobacteria (*R^2^* = 0.317 *p* = 0.004) and WS3 (*R^2^* = 0.281, *p* = 0.008) responded to χlf in the opposite direction, being more abundant in sites with lower χlf values (Appendix A).

We also analyzed the relation between the IRB and χlf in correlation assay. IRB are a group of microorganisms that could reduce amorphous Fe (III) oxides under anaerobic conditions [28,29]. We chose the reads classified as IRB in the literature [28,29] to analyze the relationship between χlf and IRB community composition. Differences in χlf between sediment samples showed a close relationship to the relative abundance of total identified IRB (Figure 2A). Regression analysis showed a linear relationship between χlf and the relative abundances of total IRB (*R^2^* = 0.331, *p* = 0.003) (Figure 2B).

The TOC was also positively correlated with Gammaproteobacteria (*r^2^* = 0.266, *p* = 0.009) shown by the Mantel test (Appendix A).

### 3.5. Effects of Geographic Distance on Microbial Community Composition

Regression analysis revealed that the relationship between geographic distance and Bray-Curtis dissimilarity had a relatively weak distance–decay pattern (*R^2^* = 0.014, *p* < 0.001) (Figure 3). This showed that geographic distance was correlated with community composition though the contribution was low. In all measured environmental variables, only pH content was spatially correlated with microbial community composition (Mantel tests, Appendix A).

### 3.6. Variation Partitioning of Microbial Community Composition

Variation partitioning analysis showed that the environmental and spatial variables could explain 24.51% of the total variation in the microbial community composition (Figure 4). Sediment environmental variables explained 17.79%, among them pure environmental variables explained 9.80 ± 1.04% of the variation (*p* < 0.001), which was higher than pure spatial variables (6.72 ± 1.08%, *p* < 0.05). Approximately 7.99% of the variation was attributed to spatially structured environmental variation (the fraction jointly explained by environmental and spatial factors). The residual 75.49% of the variation was unexplained by variation partitioning.

### 3.7. The Effect of Magnetite Center Exposure on the EET Process

A previous study of the sedimentary magnetite distribution in the gulf of Bohai Sea showed a high content of magnetite exposure at the site of 120.4° E, 39° N [8] (Figure 5A). To determine whether distance from the center of the magnetite influenced the microbial community structure, we analyzed the relationship between the relative abundances of the top 15 genera from the two cross-section samples (Figure 5A,B) and the distances from these samples to the center of magnetite. Five genera, including *Lactococcus* (*R^2^* = 0.220, *p* = 0.043), *Caulobacter* (*R^2^* = 0.217, *p* = 0.045), *Gillisia* (*R^2^* = 0.366, *p* = 0.006), *Clostridium* (*R^2^* = 0.372, *p* = 0.006), and *Sphingomonas* (*R^2^* = 0.255, *p* = 0.027) (Table 3) were found correlated with the distances from these samples to the center of magnetite. This suggests that there might be a nonrandom distribution in a horizontal orientation. The distribution of these five genera in the horizontal direction was related to the distance from the magnetite center. We speculate that the presence of magnetite produces a certain magnetic field that changes the community structure of some microbes such as DIRB. We randomly selected two sites (R3 and R5) from outside the magnetite center (R3 > R5, distance to center of magnetite) (Figure 5A), and cultured the sediments of R3 and R5 with amorphous iron (Figure 5B). The results showed that the Fe (II) content produced by the reduction of amorphous iron by the R3 culture was 7.69 mM, and the Fe (II) content produced by the R5 culture was 5.28 mM. These two enrichments were used to perform microbial fuel cells (MFC) (Figure 5C). The reactor inoculated with microorganisms from the R3 enrichment culture showed a high peak voltage of 4.02 mV. Low peak voltage of 1.99 mV was observed in the reactor inoculated with R5 enrichment culture (Figure 5D). The peak voltage of R3 was higher than that of R5. It is consistent with the distance changes of R3 and R5 to point (120.4° E, 39° N). This result showed that in the Bohai Sea sediments the closer to the magnetite center, the stronger the ability of the community to facilitate extracellular electron transfer was. This suggests that the magnetic field strength may affect the electroactivity of microbes in enrichment.

## 4. Discussion

### 4.1. Environmental Variation Plays an Important Role in Shaping Bacterial α-Diversity

In our study, the microbial α-diversity was highly correlated with salinity, DO levels, and total iron content. Corroborating our findings that salinity may shape microbial diversity, a previous study showed that increased salinity was reflected by decreased microbial activity in the surface sediments of the Qinghai-Tibetan lakes [30]. DO was also documented to be the primary driving force in mine drainage habitats, related to metabolism associated with oxygen [31], and DO may play a crucial role in taxonomic diversity on a small (vertical) scale [32].

### 4.2. Environmental Variation Influences the Microbial Community Composition

In this study, the bacterial community composition was correlated with environmental factors and was affected by salinity, pH, DO, χlf, Fe, and TOC (Figure 1E and Table 2). These findings suggest that environmental variables may be crucial to shaping the microbial community composition.

Salinity is well known to be a major contributor to microbial community structure and function [33]. The important role of salinity in bacterial communities has been found globally in heterogeneous environments and in the sedimentary ecosystems of Hypersaline Laguna Tebenquiche [34]. Consistent with those findings, we observed the most significant correlation between salinity and bacterial community structure (Table 2), because salinity is related to osmotic pressure, which changes the intracellular membrane structure and affects metabolic pathways [33,35]. Previous studies have also shown that bacterial communities in transition zones vary geographically owing to sharp salinity gradients [36]. A possible reason for the influence of salinity on bacterial distribution found in our current study is that our research sites focused on bacterial communities distributed in a geographic area with a wide salinity gradient, including estuaries, coastal margins, and open sea, where salinity values differ significantly.

χlf was correlated with community composition of many phyla, including Alphaproteobacteia, Betaproteobacteia, Planctomycetes, and Cyanobacteria. Cyanobacteria showed a positive correlation with χlf, suggesting that χlf might be related to the growth patterns of ecologically important species such as Cyanobacteria, one of the main participants of primary productivity in the global carbon cycle. In sediment, available iron is momentous for Cyanobacteria growth, and Fe oxide minerals have the largest release potential [37]. Therefore, the growth of Cyanobacteria is heavily influenced by Fe availability in all water bodies [38]. The organic matter production and the carbon cycle are also affected by Fe availability. The χlf values are coupled to iron-reducing bacterial activity in hydrocarbon contaminated sediments [39]. In our study, we found that iron-reducing bacteria were more abundant in sites with higher χlf values (Figure 2). A negative relationship between TOC and χlf was observed in the RDA assay, indicating that consumed TOC might be used for iron-reducing bacteria growth, because numerous bacteria including iron-reducing bacteria are dependent on organic matter produced by Cyanobacteria. In the Bohai Sea organic matter could associate microbial iron related metabolism with microbial-driven changes in magnetic susceptibility [40] as shown by measurable χlf values (Figure 6).

Sediment DO and water pH were found to be critical in shaping microbial community structure (Figure 1E; Table 2). The top 10 phyla in this region included Proteobacteria, Chloroflexi, Planctomycetes, and Cyanobacteia (Figure 1C), which represent the predominant phyla in the sediments of the eastern Mediterranean Sea [41]. In this study, Alphaproteobacteria, Betaproteobacteria, Planctomycetes, and Cyanobacteria increased with elevated DO and χlf (Appendix A) and decreased with increased pH (Appendix A). This suggests that Betaproteobacteria and Cyanobacteria in the tested sedimentary regions may prefer shallow-estuary sediment (low salinity, high DO, and low pH) with high χlf. pH is one of the most important factors influencing microbial energy respiration, physiology, and growth. The intracellular pH was relatively stable, and the extracellular pH depended on the level of cell metabolism. Previous reports confirmed that under low pH and high DO conditions in coal mining-associated lakes, there were high concentrations of Fe (II) and protons because of the oxidation of pyrite in mine tailings [42,43]. The produced Fe (II) was then oxidized by ferroxidans in Betaproteobacteria and precipitated as Fe (III) hydroxysulfate to the sediment [43]. In this study, we found that Betaproteobacteria prefer this kind of environment (low pH, high DO, and high χlf) and this might be related to Fe (II) oxidization.

The RDA assay showed that TOC was positively associated with pH, possibly because surface sediments contain a higher proportion of labile algal derived aliphatic organic matter and more anions [44,45]. In our study, Proteobacteria was the most abundant group of bacteria. In surface sediments that contain higher proportions of organic matter, Proteobacteria and Bacteroidetes are often prominently detected during the initial degradation of algal derived organic matter in marine sediments [44,45]. The dominant members of Bacteroidetes in the surface sediments were consistently enriched, similar to reports from previous studies [45,46]. Because Cyanobacteria is documented to be the main participant and contributor to productivity of the global carbon cycle [47,48], the occurrence of Bacteroidetes and Cyanobacteria in the surface sediment suggests that Bacteroidetes may survive better in areas rich in fresh organic matter.

Deltaproteobacteria and Gamma proteobacteria declined with increased DO and increased with elevated pH (Appendix A). This suggests that Deltaproteobacteria and Gamma proteobacteria might prefer to inhabit deep-marine sediment environments (high salinity, low DO, and high pH). Deltaproteobacteria include many IRB, such as *Geobacter*, *Anaeromyxobacter*, *Desulfobulbus*, *Desulfobacter*, *Desulfuromonas*, *Desulfuromusa*, and *Pelobacter*. In high pH and low DO environment, the increased pH changes the surface charge of the trivalent iron oxide [49]. The surface organic matter is negatively charged and released and reduction of trivalent iron might occur [49]. These results suggest that sediment salinity, pH, DO, and χlf could be good predictors of bacterial community composition variation.

It should be noted that in our study, the correlation analysis with other environmental variables was based on the relative abundance of bacterial community. The relative abundance was not independent data and reflected the mutual restriction between different taxa.

### 4.3. Environmental Variables Play a More Important Role Than Dispersal Limitation (Spatial Variables) in Conditioning Bacterial Biogeography

Of all measured environmental variables, only pH was correlated with geo-distance (Mantel tests, Appendix A). Other drivers (salinity, DO, and χlf) were not significantly correlated spatially. These results indicated that most local environmental conditions were not shaped spatially.

Despite the fact that the magnetite content in the center of the Bohai basin was high, the magnetic susceptibility did not show a correlation with this distance from the magnetite center. One possible explanation is that the sedimentary settling happens in a vertical manner from surface to deep layer, whereas magnetic susceptibility is determined as iron minerals form based on location on earth, in relation to magnetic north. Among the top 15 most dominant genera (Figure 5B), five genera (*Lactococcus*, *Clostridium*, *Caulobacter*, *Gillisia* and *Sphingomonas*) showed a clear correlation with the distance from the center of the magnetite (Table 3). This implies that the exposure of magnetite may shape the geographical distribution of these genera, and most likely by affecting the iron-related geochemical cycle these genera participate in (Figure 7). It is reported that *Lactococcus* participates in Fe (III) reduction during the external electron transfer mediated by sodium anthraquinone-2,6-disulphonate (AQDS) [50]. It is possible that *Lactococcus* sp. uses a very small portion of regenerated reducing power NADH for the reduction of external electron acceptor Fe (III) to Fe (II) in anaerobic lactic acid fermentation [51]. The lactic acid produced by *Lactococcus* was then used by *Clostridium* for Fe (III) reduction, because *Clostridium* could act as a lactic acid fermenter and Fe reducer [52]. *Caulobacter* is known to participate in metal oxidation through the biosorption and metabolism of iron [53]. That is one of the reasons that the *Caulobacter* distribution is associated with the distance from the center of the magnetite. *Sphingomonas* was identified as a microcystin-degrading bacterium during the decay of Cyanobacteria [54]. Cyanobacteria growth needed available iron [37]. Distribution of *Sphingomonas* might be adjusted according to iron presence indirectly because of its correlation with Cyanobacteria. *Gillisia* was detected as a siderophore producer in seawater or sand samples [55]. Siderophores are the metal-chelating agents that primarily function to capture the insoluble ferric iron from different habitats [55]. Numerous bacteria cannot produce siderophores but have siderophore acceptors [55]. *Gillisia* might assist other bacteria such as Cyanobacteria, *Lactococcus*, *Clostridium,* and *Caulobacter* that do not have siderophore generation capability for iron release or absorption from Fe-containing minerals. Therefore, the geographical distribution of *Gillisia* also was impacted by the presence of magnetite (Table 3).

Environmental variables explained 9.80 ± 1.04% (*p* < 0.001) of the total microbial community composition variation, which was higher than spatial factors 6.72% (*p* < 0.05) in a variation partitioning analysis (Figure 4). This further suggests that environmental variables play a more important role than spatial variables in shaping the bacterial composition and distribution. However, the environmental and spatial factors only explained about 17% of the composition variation. It would be interesting to further identify other possible factors, such as human influence, on the microbial community composition.

It has been reported that both the environmental and spatial variables play significant roles in influencing the biogeography of total microeukaryotic communities [56]. A different study showed that spatial distance (dispersal limitation) contributed more to bacterial community variation than any other factor [57]. In our current study, we showed that environmental variables were more important than the spatial variable for governing bacterial community turnover.

In the present study, OTU patterns and the bacterial community composition were correlated with geographical distance with statistically significant (*p* < 0.001) but low contribution (Figure 3). The results of the distance–decay pattern indicated that dispersal limitation may be another influential factor driving microbial biogeography. Dispersal could eliminate the distance–decay relationship by counteracting microbial compositional differentiation [58]. Limited dispersal should strengthen the distance–decay relationship [58], and the strength of correlation between dispersal limitation and microbial community composition relies on geographical distance [59] and organism size [60]. Limitations of microbial dispersal have been demonstrated at large [61] or intermediate (10–3000 km) spatial scales [59]. Dispersal limitation may exist in intermediate spatial scale at the Bohai sea (approximately 100 km). Strong dispersal limitations are often associated with increased bacterial size [62]. The bacteria found in the current study were within a relatively narrow size range, from 0.5–5 µm [63]. This could help explain why the distance–decay curve inclined slightly, which was evidence of community variation purely constrained by spatial factors (6.72%) (Figure 4). This demonstrated that dispersal limitation was associated with microbial community composition, but was not the dominant factor in shaping microbial biogeography in the Bohai Sea.

A large unexplained fraction (75.49%) was in the variation partitioning analysis. This could be potentially explained by unmeasured environmental variables, local artificial effects, and other factors.

## 5. Conclusions

Our study provides novel information about key environmental variables that influence microbial distribution and community structure in a typical coastal area. Among these variables, salinity, iron content, and χlf were crucial for bacterial α-diversity and community composition in riverine and marine surface sediment around and in the Bohai Sea. Environmental factors (explained 9.80%) played a larger role than spatial variables (explained 6.72%) in conditioning the bacterial community composition. Exposure to the magnetite center may shape geographical distribution of five DIRB. It may be caused by the interaction between microbial extracellular electrons and iron.

## Figures and Tables

**Figure 1 microorganisms-10-00006-f001:**
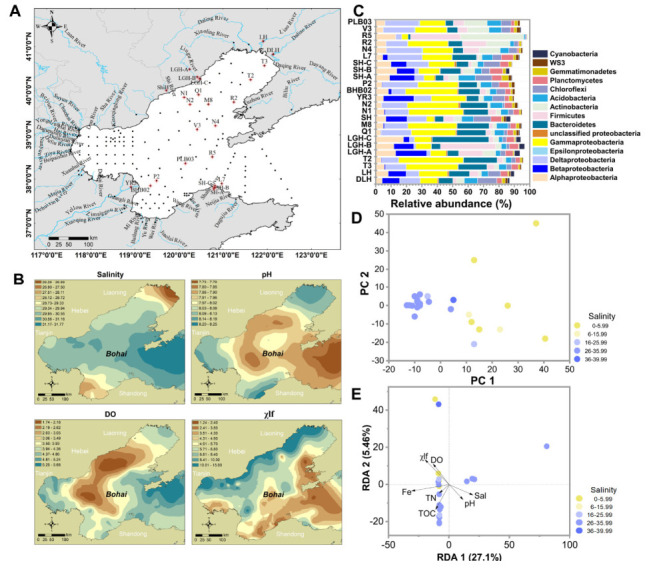
Study area, environmental variables, and microbial community composition. (**A**) A map of the study area. Black dots denote sampling sites. The letters denote site names. Red “⊕” indicates sites chosen for further study of microbial community structure, including Daliao River sites (DLH), Liao River (LH), Liugu River sites (LGH-A, LGH-B, and LGH-C), Shi River site (ShiH), Yellow River site (YR), Sha River sites (SH-A, SH-B, and SH-C), and sample sites located in the Bohai Sea (Q1, M8, T3, T2, N1, N2, L7, BHB02, PLB03, V3, N4, R2, R3, and R5). (**B**) Maps showing salinity, dissolved oxygen (DO), low-frequency magnetic susceptibility (χlf) of the sediment, and pH of the overlying water across the Bohai Sea. The maps were constructed using ArcGIS 10.0 software. (**C**) Microbial community composition in the study area at the phylum level (Proteobacteria showed by classes). (**D**) Principal component analysis (PCA) plot based on an OTU-based Bray-Curtis dissimilarity metric derived from the freshwater and marine sediment samples. (**E**) Redundancy analysis (RDA) plot of the bacterial communities and the main environmental characteristics.

**Figure 2 microorganisms-10-00006-f002:**
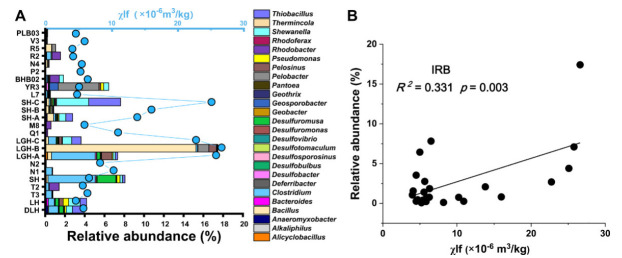
Relationship between iron reducing bacteria (IRB) and low-frequency susceptibility (χlf) values. (**A**) Relationship between genera of IRB and χlf values of sedimental samples. Sediment χlf values are reported in Appendix A. (**B**) Regression analysis between the relative abundance of identified IRB genera and χlf.

**Figure 3 microorganisms-10-00006-f003:**
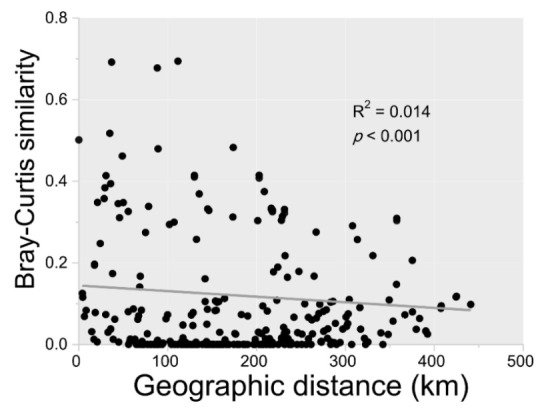
Relationship between the Bray-Curtis similarity of the microbial community and geographic distance between sampling stations. The solid line indicates the fit between geographic distance and Bray-Curtis similarity.

**Figure 4 microorganisms-10-00006-f004:**
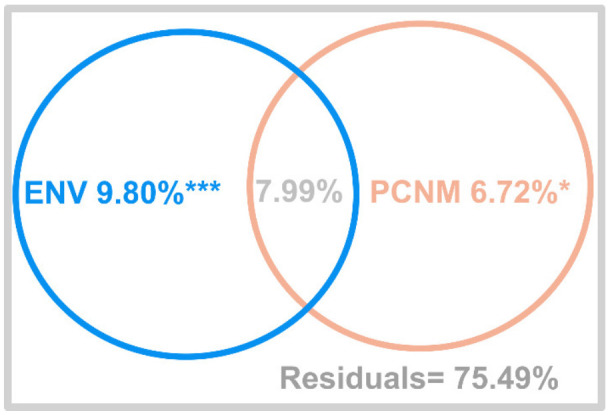
Variation partitioning of bacterial community composition in coastal sediments. The explanatory power of the pure and shared fractions of environment (ENV), and spatial factors (PCNM variables) are indicated as adjusted *R*^2^. Analysis of Variance (ANOVA) tests were carried out on the variation explained by the pure fraction. ***: *p* < 0.001; *: *p* < 0.05.

**Figure 5 microorganisms-10-00006-f005:**
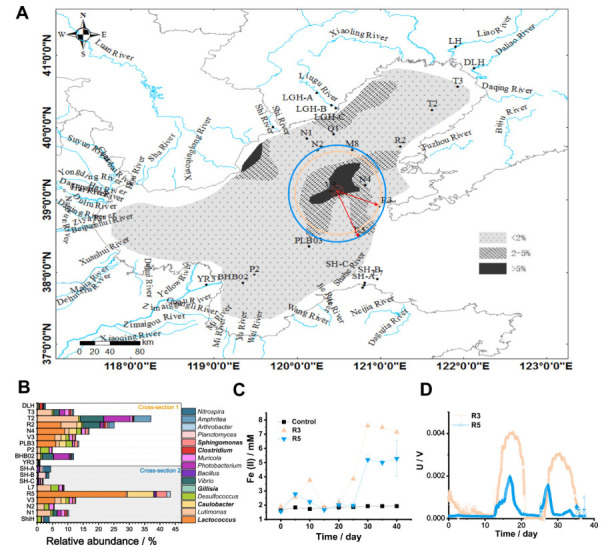
Presence of magnetite accumulation center affected dissimilatory iron reducing microorganism (DIRB) distribution and extracellular electron transfer process. (**A**) Maps showing sediment sample sites from two standard oceanic sections through the Bohai Sea, section 1 (DLH, T3, T2, R2, N4, V3, PLB3, P2, BHB02, YR3) and section 2 (SH-A, SH-B, SH-C, L7, R5, V3, N2, N1, SH). Central area of magnetite exposure (red circle) (120.4° E, 39° N) and percentage of magnetite showed in shadow. R3 and R5 with different horizontal distances (red arrows). (**B**) Relative abundances of the top 15 bacterial taxa at genus level of sample sites from two standard oceanic sections. Bold indicates the significant difference (*p* < 0.05) tested by correlation analysis between genera and geographical distance to the center of magnetite in Table 3. (**C**) Production of iron (II) in sediments of sample sites in R3 and R5 cultured with amorphous iron. (**D**) Voltage-time curves of air-cathode MFCs based on enrichment culture from R3 and R5. An external resistor of 1000 Ω was loaded between the anode and cathode. Three reactors were operated for each inoculum and the experiments were repeated three times.

**Figure 6 microorganisms-10-00006-f006:**
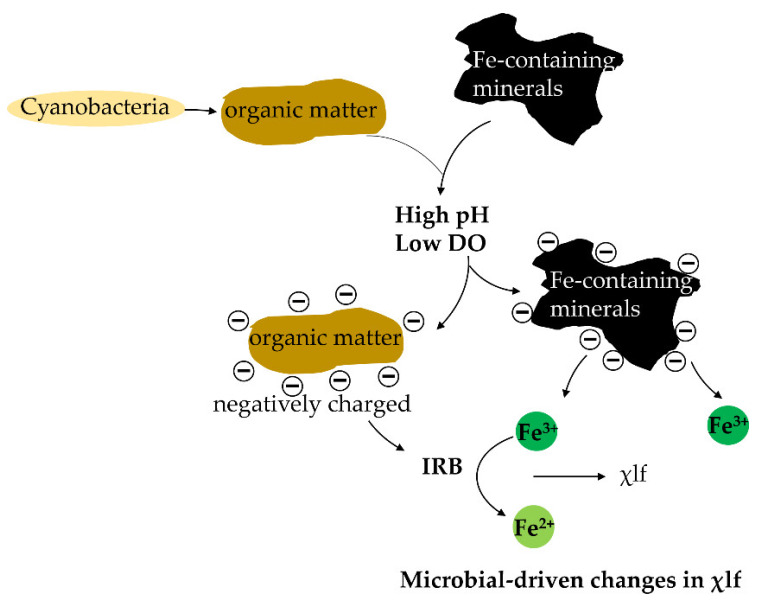
Iron reducing bacteria (IRB)-driven changes in low-frequency susceptibility (χlf). DO, dissolved oxygen.

**Figure 7 microorganisms-10-00006-f007:**
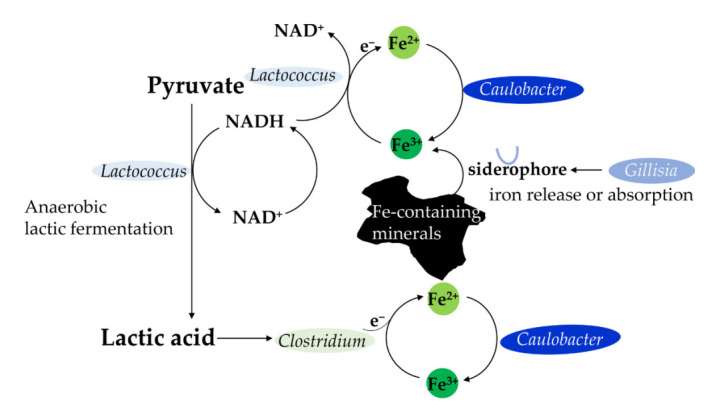
Iron-related geochemical cycle several dissimilatory iron reducing microorganism (DIRB) genera participate in.

**Table 1 microorganisms-10-00006-t001:** Relationship between chemical properties and bacterial α-diversity across samples of sediment.

	Chao 1 Index		Observed−OTU Richness (S)		Shannon Wiener Index (H’)	
	*r*	*p*	*r*	*p*	*r*	*p*
Salinity	0.202	0.015	0.230	0.009	0.091	0.257
pH	0.002	0.384	−0.007	0.432	−0.097	0.778
DO	0.063	0.187	0.136	0.084	0.007	0.365
Fe	0.104	0.127	0.229	0.027	0.373	0.046
χlf	−0.019	0.510	−0.040	0.572	−0.068	0.608
TOC	−0.075	0.837	−0.109	0.910	−0.109	0.859
TN	0.031	0.312	−0.007	0.456	−0.055	0.546

DO, dissolved oxygen; χlf, low-frequency susceptibility; TOC, total organic carbon; TN, total nitrogen. Correlation (*r*) and *p*-value were tested by correlation analysis. Bold numbers indicate significant difference (*p* < 0.05).

**Table 2 microorganisms-10-00006-t002:** Results from the PERMANOVA (adonis) model based on Bray-Curtis dissimilarity with 999 permutations.

**Variables**	***R*^2^ (Adonis)**	** *p* **
Salinity	0.161	0.001
pH	0.108	0.004
DO	0.170	0.001
Fe	0.064	0.132
χlf	0.093	0.020
TOC	0.079	0.061
TN	0.049	0.327

DO, dissolved oxygen; χlf, low-frequency susceptibility; TOC, total organic carbon; TN, total nitrogen. *p* < 0.05 indicates significant difference.

**Table 3 microorganisms-10-00006-t003:** Regression analysis between relative abundances of bacterial genera and geographical distance to the center of magnetite from 2 sections.

Genera	*R^2^*	*p*
*Lactococcus*	0.220	0.043
*Lutimonas*	0.022	0.547
*Caulobacter*	0.217	0.045
*Desulfococcus*	0.074	0.260
*Gillisia*	0.366	0.006
*Vibrio*	0.083	0.231
*Bacillus*	0.044	0.391
*Photobacterium*	0.108	0.170
*Muricola*	0.087	0.221
*Clostridium*	0.372	0.006
*Sphingomonas*	0.255	0.027
*Planctomyces*	0.045	0.382
*Arthrobacter*	0.151	0.101
*Amphritea*	0.063	0.300
*Nitrospira*	0.038	0.424

Coefficient of determination (*R^2^*) and *p* < 0.05 indicates significant difference tested by regression analysis.

## Data Availability

The datasets used and/or analyzed during the current study are available from the corresponding author on reasonable request.

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
