# Peer review of "Effects of Magnetic Minerals Exposure and Microbial Responses in Surface Sediment across the Bohai Sea"

_microorganisms, 2021, doi:10.3390/microorganisms10010006_

Round 1
Reviewer 1 Report
Dear Authors,
Your manuscript concerns an important issue, connected with environment pollution by magnetic wastes and provide a novel information about key environmental variables that influence microbial distribution and community structure in coastal area. I have some suggestions how improve the manuscript:
Conclusions should better reflect the obtained results and objectives of the work.
Please correct the numeration of subsections, remove the additional dots.
Line 54
Use a proper indexes in chemical formulas.
Lines 86, 289, 351
Figures 1 and 2, 5b, - figure legends are unreadable.
Line 94
Probacteria or Proteobacteria?
Author Response
Response to Reviewer 1 Comments
Dear Authors,
Your manuscript concerns an important issue, connected with environment pollution by magnetic wastes and provide a novel information about key environmental variables that influence microbial distribution and community structure in coastal area. I have some suggestions how improve the manuscript:
- Conclusions should better reflect the obtained results and objectives of the work.
Response: The conclusion has been revised.
- Please correct the numeration of subsections, remove the additional dots.
Response: Deleted.
- Line 54 Use a proper indexes in chemical formulas.
Response: Corrected.
- Lines 86, 289, 351 Figures 1 and 2, 5b, - figure legends are unreadable.
Response: Revised.
- Line 94 Probacteria or Proteobacteria?
Response: Corrected.
Reviewer 2 Report
The manuscript presents and discusses a multitude of experimental results concerning the influence of environmental factors including iron magnetic minerals on the microbial community of Bohai Sea, China.
The manuscript deserves publication, but only after the authors will reply to some comments which can be found on the annotated manuscript. I would suggest paying more attention to the remained 83% of the variability of microbial communities not explained within present models.
NB The Conclusion section should be significantly enlarged.

Author Response
Response to Reviewer 2 Comments
- add ORCID ID if exist
Response: We do not currently have ORCID ID.
2) +/- uncertainty would increase the quality of manuscript
Response: Added.
3) I would suggest the manuscript to be revised by a native English speaking
Response: The manuscript has been reviewed and edited by a native English speaking scientist.
4) In Figure 1, please us greater fonts ar split figure so as the legend to be readable. I would suggest to use 300 dpi .tif file. jpg file contain artifacts due to compression
Response: This has been revised as suggested.
5) please explain OTU operational taxonomic units
Response: This has been added.
6) in the final variant please use widow-orphan option
Response: The “Window-orphan option” has been enabled.
7)In Table 1, r^3 ???
r = 0.202 is a weak correlation
sqrt(0.202) = 0.587 is also not a strong correletion
could you explain why you used r^3 instead r.
Response: The “3” is the note of the table. It has been deleted in the newly submitted manuscript to prevent confusion.
8) In figure 2A, please use a greater fonts or a greater figure.
Response: Done.
9) In figure 2B, two remarks:
- explain why you have used r^2 instead of r as defined by Pearson ?- are you sure all data are normally distributed to use the Pearson's coefficient. my be Spearman's rho would most appropriate?
Response: We apologize for the mistake and confusion. Regression analysis, instead of Pearson correlation is used here. The original text has been corrected.
10) In figure 3, either at p <0.001, r = 0.118 shows a very weak correletion
Response: We agree with the reviewer. This has been indicated in the revised manuscript.
11) In table 3, r or r^2 or r^3 ????
Response: Corrected.
12) In discussion, 9.8 is of course greater than 6.8, but all together explain about 17% of total composition variation.
Response: We agree with the reviewer. Other possible factors human influences, have been briefly discussed.
13)In the paragraph “One limitation of the study is that the temporal ~~”, these are presumptions, which should be explained in future experiments
Response: This paragraph has been deleted.
14) In Supplementary Materials, all of them could be included in Appendices. MDPI journals have not limited number of pages
Response: The supplementary materials have been included as appendices of the revised manuscript.
Reviewer 3 Report
The paper studies the effects of magnetic minerals on the community of betterspresent in the sediments of the seabed near the estuaries of many rivers.
The paper is interesting but I point out some flaws.
1)The acronyms of the parameters are not well defined and therefore it is very
difficult to follow the subsequent discussion;
2)All discussion should be summarized in tables and graphs in order to be
more effectively understood
3)The conclusions are too concise and do not help to summarize the whole work
4)The number of references is excessive. Eliminate older references where possible
5)" Magnetic suceptibility (mainly low-frequency susceptibility values, χlf) of samples from the sites 200
across the Bohai Sea were measured" What exactly does this mean?
What did the authors measure?
The low and high frequency magnetic susceptibility can give information
on the size of the magnetic grains and the comparison between the two parameters can
give information on the origin of the magnetic materials. The authors explain what they measured
Author Response
Response to Reviewer 3 Comments
The paper studies the effects of magnetic minerals on the community of betters
present in the sediments of the seabed near the estuaries of many rivers.
The paper is interesting but I point out some flaws.
1)The acronyms of the parameters are not well defined and therefore it is very
difficult to follow the subsequent discussion;
Response: All acronyms have been carefully defined.
2)All discussion should be summarized in tables and graphs in order to be
more effectively understood
Response: Thank you for your suggestions. Two graphs (Figures 6 and 7) have been added to the revised manuscript.
3)The conclusions are too concise and do not help to summarize the whole work
Response: The conclusion has been revised.
4)The number of references is excessive. Eliminate older references where possible
Response: We have carefully checked and deleted some of the non-essential references.
5)" Magnetic suceptibility (mainly low-frequency susceptibility values, χlf) of samples from the sites 200 across the Bohai Sea were measured" What exactly does this mean? What did the authors measure? The low and high frequency magnetic susceptibility can give information on the size of the magnetic grains and the comparison between the two parameters can give information on the origin of the magnetic materials. The authors explain what they measured.
Response: We apologize for the vagueness of the wording. Only the low frequency susceptibility was measured. This has been corrected.
Round 2
Reviewer 2 Report
In my opinion it is now OK.
Author Response
Thank you.
Reviewer 3 Report
the authors have addressed all my concerns and now the document is ready for publicationAuthor Response
Thank you.